# Diaphragmatic Liver Herniation after Radiofrequency Ablation of a Secondary Liver Tumor

**DOI:** 10.3390/diagnostics14010026

**Published:** 2023-12-22

**Authors:** David Hoskovec, Josef Hořejš, Zdeněk Krška, Soňa Argalácsová, Pavol Klobušický

**Affiliations:** 11st Department of Surgery, 1st Faculty of Medicine Charles University and General University Hospital, 121 08 Prague, Czech Republic; zdenek.krska@vfn.cz; 2Department of Radiodiagnostics, 1st Faculty of Medicine Charles University and General University Hospital, 121 08 Prague, Czech Republic; josef.horejs@vfn.cz; 3Department of Oncology, 1st Faculty of Medicine Charles University and General University Hospital, 121 08 Prague, Czech Republic; sona.argalacsova@vfn.cz; 41st Medical Faculty, Charles University, 128 00 Prague, Czech Republic; klobusicky@icloud.com

**Keywords:** diaphragm hernia, radiofrequency ablation, cancer, complication, RFA

## Abstract

Radiofrequency thermal ablation (RFA) is widely used and has been accepted for the treatment of unresectable tumors. The leading technique that is used is percutaneous RFA under CT or US guidance. Multicenter surveys report acceptable morbidity and mortality rates for RFA. The mortality rate ranges from 0.1% to 0.5%, the major complication rate ranges from 2% to 3%. Diaphragmatic injury is a rare complication and it is described after RFA of subdiaphragmatic tumors. Most of them are without clinical importance. There are some case reports about diaphragmatic herniation of the intestine into the pleural cavity. We present a case of diaphragmatic perforation resulting in the herniation of the liver into the pleural cavity. A thoracotomy was performed, the liver was lowered back into the peritoneal cavity and the perforation was closed with mesh.

The liver is a common site for both primary and secondary tumors. The most common are hepatocellular carcinoma and colorectal cancer metastasis. Surgery is the standard treatment for both malignancies but only 5% to 15% of patients are candidates for curative surgery. RFA is used for patients who are unable to be operated on. During RFA, the target tissue is heated, and the malignant cells vanish. The energy is produced by a very high frequency current generator (200 to 1200 KHz). The effect of radiofrequency waves was described by d’Aersonvai in 1891. His observation led to the development of medical devices in the 1990s. Liver tumors have been treated since 1993 (Rossi). RFA is used in the Czech Republic since 2000 (Horejs).

The complications of RFA could be classified according to various points of view. One way is to sort the difficulties as related to probe placement (bleeding, infection, tumor seeding), related to thermal energy (nontarget thermal injury, skin burns) and related to specific complications of the targeted organ (organ failure) [1]. Another possibility is division according to the severity of the complication (Proposal of the International Working Group of Image-guided Tumor Ablation) [2,3]:

Major—if untreated, it might threaten the patient’s life, lead to substantial morbidity and disability, result in hospital admission or prolong hospital stay.

Minor—all other complications

Side effects—are not real complications. They are expected but unwanted consequences of RFA (for example pain or temperature not higher than 38 degrees Celsius).

A 47 year old woman was operated on due to breast carcinoma 10 years ago. A left mastectomy and axillar lymphadenectomy were performed (2000). Mammoplasty and thoracic wall reconstruction were performed 2 years later (2002). Three metastases in the right liver lobe were diagnosed 7 years after the primary surgery (2007). The biggest metastasis was 8 cm × 7 cm × 7 cm, the other two had a diameter of about 1 cm (Figure 1). The diagnosis was confirmed by a CT navigated biopsy. The pathological examination was ductal breast carcinoma (estrogen receptor in nearly all cells, progesterone receptor in 60%). The initial treatment was chemotherapy as a result of which two metastases disappeared and the last one has shrank to 50 mm × 30 mm × 30 mm.

The last one was destroyed by RFA (2008) (Figure 2). RFA (RITA Medical system AngioDynamics) was performed under CT guidance in two cycles (March and April 2008). The StarBurstTM XL Semi-Flex probe was used and deployed to 5 cm. Target temperature was 105 °C and power was 150 W. Immediately after the procedure, the patient was without any clinical or paraclinical findings of complications.

She was regularly checked by an oncologist during the follow up. 15 months after the second RFA, some diaphragmatic elevation was found. But the patient was without clinical problems (Figure 3).

She started to suffer from breathlessness one year later. CT examination showed a diaphragmatic hernia which contended the right liver lobe and part of the large bowel (Figure 4 and Figure 5).

The patient was indicated for surgery. A thoracotomy was performed and the right liver lobe and transversal colon were found inside the thoracic cavity (Figure 6). The place of the previous RFA on the liver surface was clearly visible. The reduction of the hernia was difficult because the diameter of the defect was less than volume of the liver tissue above the diaphragm. It was necessary to cut the diaphragm after which the operating surgeon was able to move the liver under the diaphragm and back into the abdominal cavity. The suture of the diaphragm was covered with the mesh (Figure 7).

CT examination one month later showed normal anatomical conditions and the mesh was clearly visible (Figure 8). Multiple metastases were found 15 months after surgery (liver, lung, bones) and the patient was treated by palliative chemotherapy. She died 42 months after surgery due to generalization. There were no signs of diaphragmatic hernia recurrence.

The frequency of RFA complications is acceptable. The reported major complication rate is up to 3% and the minor complication rate is up to 9%. Mortality related to RFA is about 0.5% [2,4].

Most common complications after liver RFA are bleeding, infection (peritonitis or intrahepatic abscess), and injury of the bile duct. Liver failure is described occasionally [5,6,7].

Injury of the diaphragm is quite rare and there are usually only case reports described in the literature [8,9,10]. Most of them were found accidentally and are without clinical symptomatology. A risky position of the tumor is in the dome of the liver. Possible causes of the perforation of the diaphragm include contact between the tumor and the diaphragm. A small defect in the diaphragm could probably widen due to intraabdominal pressure and it could cause diaphragmatic herniation. The interval between RFA and diaphragmatic herniation is usually several months. It is recommended to close small defects immediately after diagnosis to prevent such herniation. A retrospective review published in 2002 found six diaphragmatic injuries after 3670 RFAs [4]. Despite the small number of perforations, three of them were fatal (liver abscess and sepsis, bronchobiliary fistula and hemorrhagic cardiac tamponade). In fact, the deaths were not only due to the diaphragmatic injury but also due to the injury of other organs. The Korean Study Group of Radiofrequency Ablation found a 0.07% rate of diaphragmatic perforation in their survey published in 2003 [7]. Head described diaphragmatic burns in 17% of all RFA treatments of tumors leaning against the diaphragm [3]. According to his study, the symptoms of diaphragmatic injury were only pain in the right shoulder and the thickening of the diaphragm. The rate of diaphragmatic hernias was not mentioned.

In our case the injury of the diaphragm probably developed at the end of the RFA because the active ablation of the liver was still in progress during the withdrawal of the probe to prevent the seeding of cancer cells. The defect widened and the right liver lobe passed through the diaphragm and into the thoracic cavity. The elasticity of the diaphragm and the elasticity of the liver tissue resulted in the volume of the liver that passed into the thoracic cavity being larger than the diameter of the defect resulting in something like a strangulation line on the liver.

RFA in skilled hands is an effective and safe method to destroy malignant tumors in the liver and other organs. Despite the fact that the rate of complications is low, it is necessary to know the specific types of complications related to the method. General recommendations to avoid complications or decrease their frequency are:–Careful patient selection;–Combination of RFA with other techniques (if possible);–Selection of guiding modality and anatomical approach;–Early detection and appropriate management of complications;–A multidisciplinary approach is a condition sine qua non in case of complications.

## Figures and Tables

**Figure 1 diagnostics-14-00026-f001:**
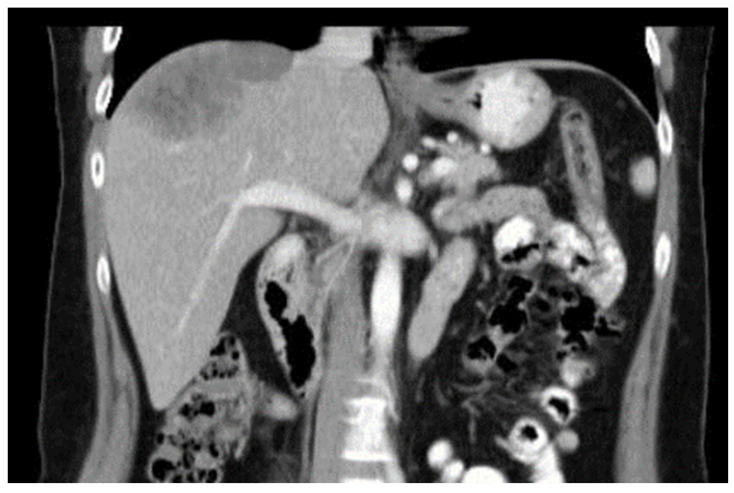
Initial finding of the liver metastasis.

**Figure 2 diagnostics-14-00026-f002:**
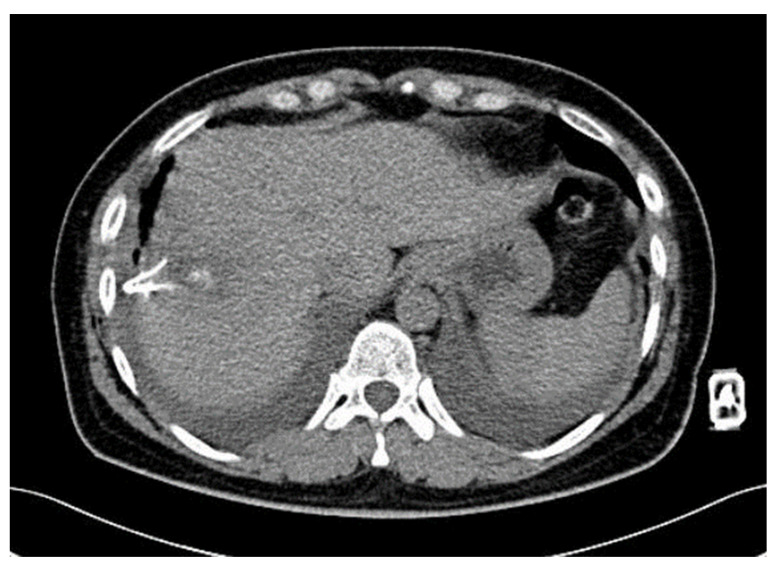
RFA probes in the liver metastasis.

**Figure 3 diagnostics-14-00026-f003:**
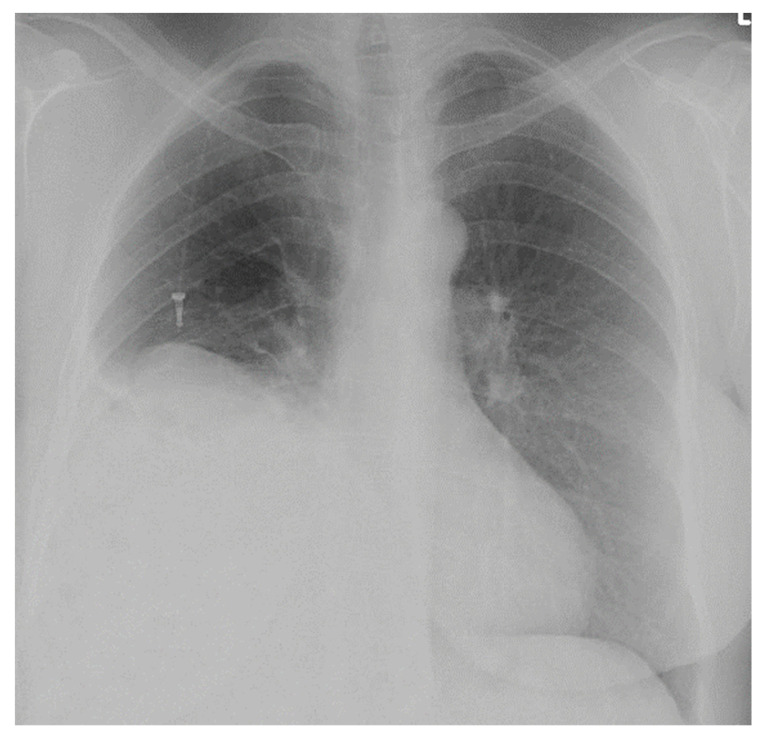
Elevation of the right diaphragm 15 months after second RFA.

**Figure 4 diagnostics-14-00026-f004:**
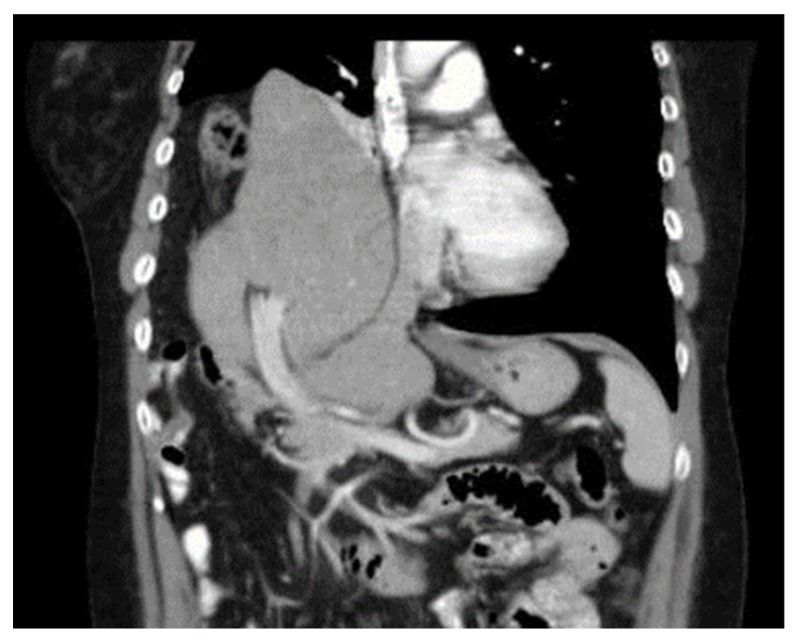
CT scan with liver in the thoracic cavity (frontal plane).

**Figure 5 diagnostics-14-00026-f005:**
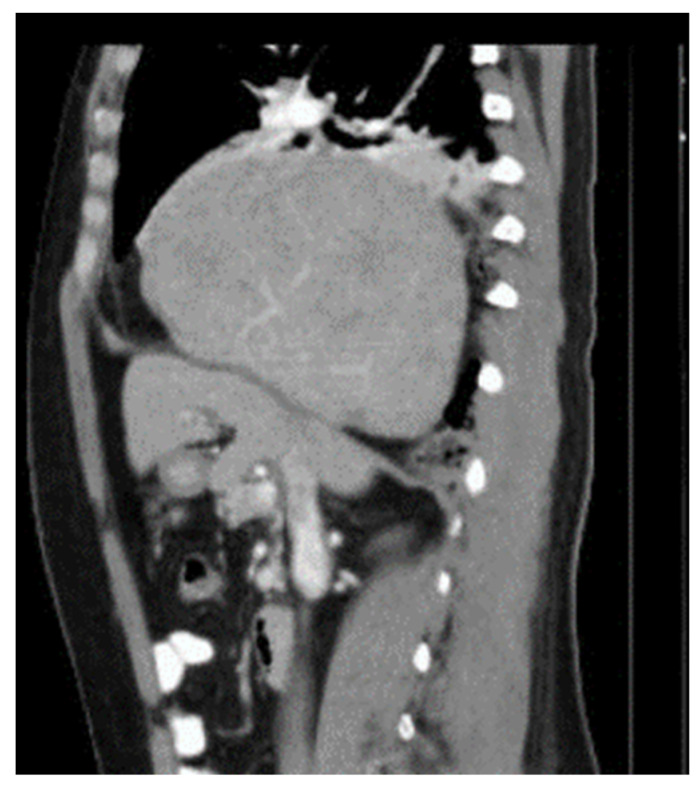
CT scan with liver in the thoracic cavity (sagittal plane).

**Figure 6 diagnostics-14-00026-f006:**
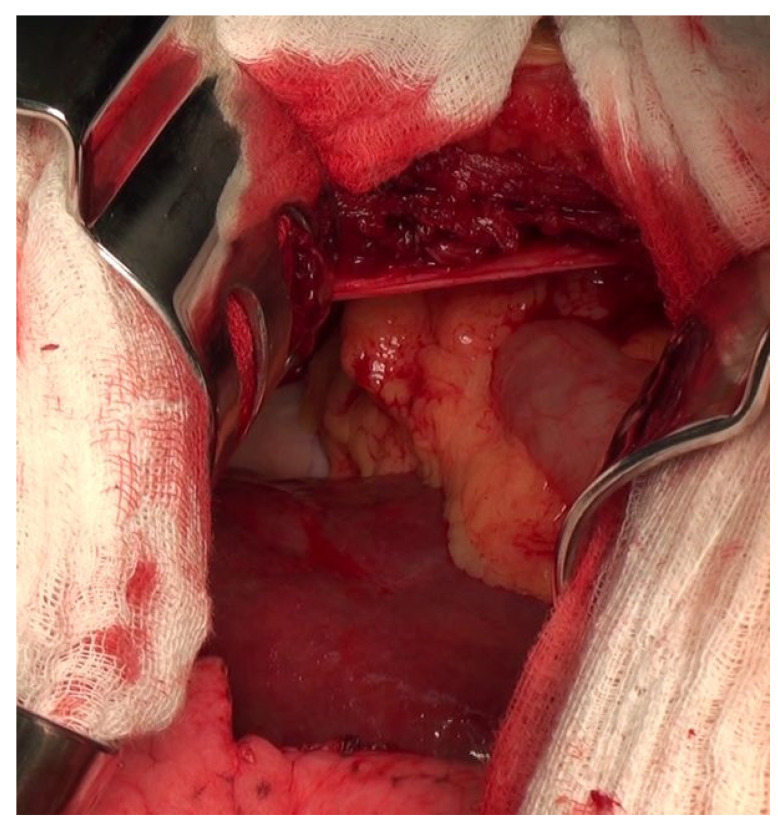
Preoperative view into thoracic cavity.

**Figure 7 diagnostics-14-00026-f007:**
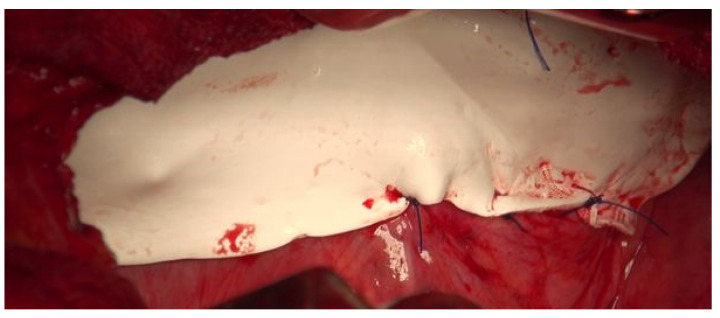
PTFE mesh covering the suture of the diaphragm.

**Figure 8 diagnostics-14-00026-f008:**
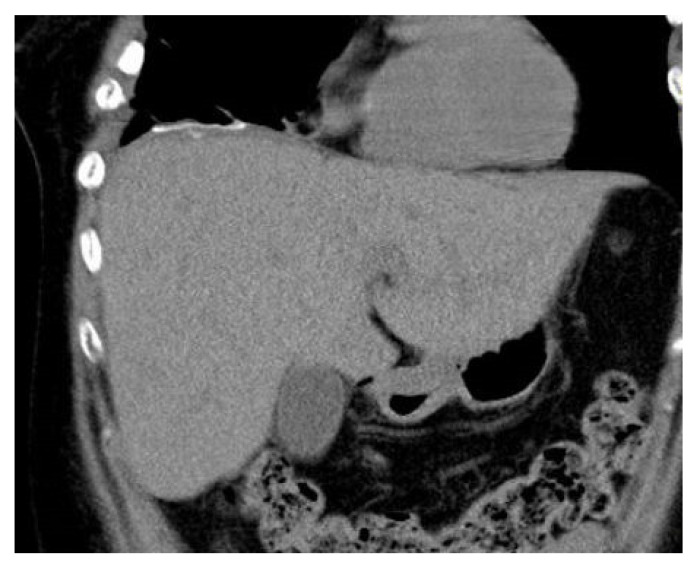
CT examination one month after surgery.

## Data Availability

Data are contained within the article.

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
