# Peer review of "Diaphragmatic Liver Herniation after Radiofrequency Ablation of a Secondary Liver Tumor"

_diagnostics, 2023, doi:10.3390/diagnostics14010026_

Round 1
Reviewer 1 Report
Comments and Suggestions for Authors
A patient with breast CA developed liver metastasis. She received chemotherapy and two RFA to treat the liver metastasis. Diaphragmatic hernia of liver developed 15 months later. The hernia was repaired by surgery successfully. The reported cases of diaphragmatic hernia after RFA are still rare.
Comments
1. Please indicate the location of the breast cancer, R or L?
2. Please shows pre-RFA and post-RFA CT images in arterial phase. Please give information about the original tumor size and post ablation necrosis size.
3. Please number of each image and describe the findings in legends.
4. Was there any cytology or histology diagnosis in liver tumors?
Author Response
Dear reviewer,
Thank you for your time and review.
All your comments have been edited and added to the text
- Please indicate the location of the breast cancer, R or L?
The tumor was on the left side
- Please shows pre-RFA and post-RFA CT images in arterial phase. Please give information about the original tumor size and post ablation necrosis size.
Unfortunately we do not have images in arterial phase in our archive. The original size of the tumor was 8 x 7 x 7 cm.
- Please number of each image and describe the findings in legends.
Done
- Was there any cytology or histology diagnosis in liver tumors?
We confirmed the diagnosis by CT navigated biopsy
Best regards and merry Christmas
DH
Reviewer 2 Report
Comments and Suggestions for Authors
This is an interesting case report with very relevant images. I suggest describing the whole case (with the facts only) and showing the imaging before discussing the issues and presenting the published literature.
Author Response
Dear reviewer,
Thank you for your time and review.
We edited the text according to your recommendation
Best regards and merry Christmas
DH